# Low Concentration of Rotenone Impairs Membrane Function of *Spodoptera litura* Cells by Promoting Their Aggregation

**Sukun Lin** [1,2,†], **Kaijie Xu** [2,†], **Qingpeng Zhang** [2], **Qiuming Zhu** [1], **Muhammad Musa Khan** [2], **Zhixiang Zhang** [2,*] and **Dongmei Cheng** [1,*]

1   Department of Plant Protection, Zhongkai University of Agriculture and Engineering,
    Guangzhou 510225, China
2   Key Laboratory of Natural Pesticide and Chemical Biology of the Ministry of Education, South China
    Agricultural University, Guangzhou 510642, China
*   Correspondence: zdsys@scau.edu.cn (Z.Z.); zkcdm@zhku.edu.cn (D.C.)
†   These authors contributed equally to this work.

**Abstract:** Rotenone is a widely used botanical insecticide, which can inhibit the mitochondrial respiratory chain of various insect cells, while the mechanism of its toxicity to insect cells needs further investigation. The purpose of this study was to understand the toxicity level of low (0.2 μg/mL) and high (20 μg/mL) concentrations of rotenone in *Spodoptera litura* cells (SL-1) using trypan blue and Coomassie brilliant blue staining assays. Our study found that rotenone possessed cytotoxicity against SL-1 cells with varying effects of action between high and low concentrations. After low-concentration rotenone treatment, the SL-1 cells showed obvious aggregation time-dependently, with the fastest aggregation rate observed under the first 8 h of treatment time, but no such phenomenon was observed at high concentration. Furthermore, this aggregation phenomenon caused SL-1 cells to squeeze each other and led to the destruction of the cell membrane structure and function. Taken together, the results suggested that treatment with a low concentration of rotenone exhibited a chronic toxic effect that was significantly different from treatment with a high concentration of rotenone, which provides new insight into the cytotoxic mechanism of rotenone.

**Keywords:** botanical insecticide; cytotoxicity; membrane structure; ovarian cell; pest management

## 1. Introduction

Botanical insecticides are natural products extracted from plants that can limit or reduce pest populations [1]. They are generally safer than synthetic pesticides for the environment as they can be quickly degraded, as well as more friendly to non-target organisms [2]. Moreover, botanical insecticides are often unique and complex, with multiple modes of action; therefore, pests are unlikely to develop resistance [3]. With the increasing potential risks of pesticides to the environment and human health, people have put forward higher requirements for pesticides [4]. In recent years, botanical pesticides have been considered attractive alternatives to synthetic chemical pesticides for pest management, and the use of botanical pesticides has been increasing globally each year [5]. Currently, the pesticide market is still dominated by synthetic chemical pesticides, while the market for botanical pesticides is relatively small, because of the limited research in this area, as well as the lack of information on application methods [6]. Hence, it is necessary to carry out more research on botanical pesticides to provide support for their application.

As an excellent botanical insecticide, rotenone is a naturally occurring toxin derived from the roots of some plants belonging to the Fabaceae family, such as *Derris elliptica* and *Lonchocarpus* [7]. Rotenone has insecticidal activity against a variety of pests and has been largely used as an insecticide in vegetable gardens for many years [8,9]. In aquaculture, rotenone is also employed to kill nuisance fish populations in lakes and reservoirs to maintain the biotic integrity [10]. Additionally, rotenone is used as a model drug for

Parkinson's disease [11]. In summary, rotenone has great application potential in plant protection, aquaculture, and medicine, among other fields. However, the current research on the mechanism of rotenone is insufficient, especially in terms of its cytotoxic mechanism in insects.

Rotenone is a well-known neurotoxicant that prevents energy production by blocking the electron transfer in the mitochondrial respiratory chain [12]. It mainly interacts with a component between NADH dehydrogenase and coenzyme Q, inhibits complex-I of the mitochondrial respiratory chain, and results in the accumulation of reactive oxygen species (ROS) causing oxidative stress, and further blocking the adenosine triphosphate (ATP) generation in organisms [13,14]. Subsequently, ATP depletion leads to insect respiratory failure, reduction in heart rate, paralysis, and ultimately death because of its insufficient energy [15,16]. Previous studies have shown that rotenone is not only toxic to some pests but also exhibits significant antifeedant activity and effective respiratory inhibition [17,18]. In addition, only a few studies have focused on the cytotoxic effects of rotenone on insects; due to its high lipophilicity, it easily crosses the biological membranes and accumulates in the cell to play a role [19], and thus, it is worthy of further research to study the insecticidal mechanism of rotenone at the cellular level. It is reported that sublethal chronic exposure to rotenone leads to the death of neurons and mitochondrial oxidative damage in *Drosophila* [20]. Sl-1 cells are the ovarian cells of *Spodoptera litura* and the model cells for toxicological research. A recent study showed that rotenone can cause the necrosis of SL-1 cells, and the depolarization of mitochondria may play a key role in rotenone-induced cell necrosis [21]. From the existing literature, there are few research perspectives on the cytotoxic effect of rotenone on pests, and more investment is still required. Therefore, the research objective of this study was to investigate the toxicity level of rotenone at low concentrations (0.2 μg/mL) to *S. litura* larvae and ovarian cells (SL-1), and to provide an important reference for a more comprehensive understanding of the mechanism of rotenone.

## 2. Materials and Methods

### 2.1. Chemicals, Insect, Cell Line, and Culture Conditions

Rotenone (98.6%) and Coomassie brilliant blue (CBB) R250 were purchased from Sigma-Aldrich Chemicals Company (St. Louis, MO, USA). Rotenone was dissolved in dimethyl sulfoxide (DMSO) as a stock solution (1000 μg/mL) and stored at 4 °C. Propidium iodide (PI) was purchased from Nanjing Key Gen Biotechnology Company (Nanjing, Jiangsu, China).

*S. litura* was collected from the vegetable field in Guangzhou, and was fed with feed for seven generations in the laboratory. The feeding conditions were temperature $24 \pm 2$ °C, relative humidity 60–80%, and photoperiod 16-8D.

The SL-1 cell line was obtained from the Key Laboratory of Natural Pesticide and Chemical Biology of the Ministry of Education, South China Agricultural University, Guangzhou, China. The SL-1 cells were cultured with Grace's insect cell culture medium containing 8% fetal bovine serum and grown in 25 cm$^2$ flasks at 27.5 °C in a humidified atmosphere of 5% $CO_2$ in the air. The cultures were subcultured every 3 days. Cells in the logarithmic phase of growth were used in all experiments at a density of $1.0 \times 10^5$ cells/mL.

### 2.2. Toxicity of Rotenone to S. litura Larvae and SL-1 Cells

The toxicity experiment of rotenone against the 2nd instar larvae of *S. litura* was determined by the method of Qin et al., and the larvae were treated with diets containing different concentrations of rotenone, respectively [22]. The mortality rate after three days of treatment was calculated using the method of Lin et al. [23]. Based on the toxicity results, using a high concentration (20 μg/mL) of rotenone as a reference point, the toxicity of rotenone at a low concentration (0.2 μg/mL) near $LC_{30}$ to SL-1 cells was studied by the trypan blue staining method. Trypan blue is often used as a cell viability dye to detect whether cells are viable; after trypan blue staining, live cells will not be stained blue, while

dead cells will be stained blue [24]. In reference to the method of Tominaga et al. [25], 2 mL of SL-1 cells at a density of $1.0 \times 10^5$ cells/mL were seeded in dishes ($\varPhi = 6$ cm) containing 0.05% DMSO and incubated for 24 h. Afterward, the medium was exposed to 0.2 and 20 μg/mL rotenone, whereas the control group was treated with 0.05% DMSO alone. After 24, 48, and 72 h, 100 μL of SL-1 cell suspension was pipetted into a centrifuge tube having 100 μL 0.4% Trypan blue staining solution respectively, and stained for 3 min. Later, a little cell suspension was spread on the hemocytometer and covered with a cover glass. The SL-1 cells were observed under an Eclipse TE2000-E inverted microscope (Nikon Co., Ltd., Tokyo, Japan) equipped with a digital camera subsequently. In order to accurately determine the toxic activity of rotenone on SL-1 cells at the above two concentrations, the number of dead cells and live cells were counted under the inverted microscope, and the cell death percent was calculated using the following equation:

$$\text{Cell death percent (\%)} = m_1/(m_1 + m_2) \times 100\%$$

where $m_1$ = the number of dead cells, and $m_2$ = the number of live cells.

### 2.3. Effects of Low Concentration of Rotenone on the Aggregation of SL-1 Cells

In the toxicity experiment, we found that SL-1 cells appeared to aggregate under 0.2 μg/mL of rotenone treatment. In order to explore this phenomenon, the aggregation rates and increments at different time periods of SL-1 cells within 24 h under 0.2 μg/mL of rotenone were measured by an inverted microscope. For instance, 10 random points were selected at each time point (0, 4, 8, 12, 16, 20, and 24 h) and pictures were taken continuously, the numbers of aggregated cells and non-aggregated cells were counted, and the cell aggregation percent and increment were calculated using the following equation:

$$\text{Cell aggregation percent (\%)} = n_1/(n_1 + n_2) \times 100\%$$

where $n_1$ = the number of aggregated cells, and $n_2$ = the number of non-aggregated cells.

$$\text{Increment of cell aggregation percent (\%)} = (p_{t+4} - p_t) \times 100\%$$

where $p_t$ = the cell aggregation percent at $\text{time}_t$, and $p_{t+4}$ = the cell aggregation percent at $\text{time}_{t+4}$.

### 2.4. Effects of Rotenone on the Skeleton of SL-1 Cells

The cytoskeleton is closely related to cell movement. Through the abnormal cell aggregation phenomenon in the above experiment, the Coomassie brilliant blue (CBB) staining method was used to further study the effects of the above two concentrations of rotenone on the skeleton of SL-1 cells. SL-1 cells, seeded at $1.0 \times 10^5$ cells/mL in dishes ($\varPhi = 6$ cm), were incubated with 0.2 and 2.0 μg/mL rotenone for 24 h and treated with 1% Triton X-100 for 15 min at room temperature. The cells were rinsed in McIlvaine buffer solution thrice and then fixed in methyl alcohol containing phosphate-buffered saline (PBS) for 10 min. After staining with 0.2% CBB R250 for 1 h, the cells were observed using the electron microscope (Olympus Co., Ltd., Tokyo, Japan).

### 2.5. Effects of Rotenone on the Morphology of SL-1 Cell

To further study the effects of the abnormal cell aggregation on SL-1 cells, a scanning electron microscope (Philips Co., Eindhoven, The Netherlands) was used to observe the morphology of SL-1 cells after being treated with the above two concentrations of rotenone. SL-1 cells, seeded at $1.0 \times 10^5$ cells/mL in dishes ($\varPhi = 6$ cm), were incubated with 0.2 and 2.0 μg/mL rotenone for 24 h, and the rotenone-treated SL-1 cells were fixed in 4% glutaraldehyde at 4 °C for 4 h and rinsed thrice in PBS. Next, fixation was carried out in 1% osmic acid for 1 h and then rinsed thrice in PBS. The cells were dehydrated twice in ethyl alcohol serial solution, subsequent to infusion in isoamyl acetate, and then dried for 3 h on

ice. Ultimately, the cells were sprayed with metal and observed under a scanning electron microscope.

### 2.6. Effects of Rotenone on the Membrane Function of SL-1 Cells

PI is a macromolecular fluorescent dye that cannot penetrate the normal cell membrane. When the cell membrane is damaged, the permeability of the cell membrane will increase, and PI can pass through the cell membrane, enter the nucleus, and combine with DNA, showing red under the fluorescence microscope, so it can be used to evaluate the changes in permeability after pesticide treatment. In the cell morphology observation, we found that under treatment with 0.2 µg/mL of rotenone, the membrane of SL-1 cells was damaged due to cell aggregation. Therefore, the permeability and stability of the cell membrane after treatment with 0.2 µg/mL of rotenone were determined. SL-1 cells were seeded and cultured as described in 2.3. At 24 h, 48 h, and 72 h after treatment, the SL-1 cells were washed twice with PBS, 10 µL of PI was added for staining for 10 min, and the remaining PI in the culture medium was washed with PBS. Stained nuclei were observed with a fluorescence microscope.

### 2.7. Data Analysis

The data in the figures are expressed as mean ± standard error from three replicates. Cell death percent, aggregation percent, and increment of cell aggregation percent were transformed to arcsine square root values and analyzed by one-way analysis of variance (ANOVA) using SPSS Statistics, Version 17.0, 2009 (International Business Machines Corporation, Armonk, NY, USA). If significant differences occurred among treatments, means were separated by Tukey's honestly significant difference (HSD) test at $p < 0.05$ level.

## 3. Results

### 3.1. Toxicity of Rotenone to S. litura Larvae and SL-1 Cells

The results of the indoor toxicity of rotenone to *S. litura* are shown in Figure S1. It can be seen that the mortality rates of different concentrations of rotenone (0.1~20 µg/mL) after 3 days of treatment were different. A mortality rate of 7.78% was evident when the larvae were treated with rotenone at 0.1 µg/mL; the mortality rates gradually increased with the concentration and reached 73.33% at the concentration of 20 µg/mL. By contrast, the mortality rate of the *S. litura* without rotenone treatment was only 5.56%, which was significantly lower than that of *S. litura* treated with rotenone (0.2 µg/mL or more). Additionally, it was found that the undead larvae appeared to be slow and weak after rotenone treatment at 0.2 µg/mL, so we speculated that there may be some sublethal effects in *S. litura* at low concentrations of rotenone.

Using a high concentration (20 µg/mL) of rotenone as a reference point, the cytotoxic effects of low concentration (0.2 µg/mL) rotenone on SL-1 cells was studied, and the results are shown in Figures 1 and 2. Significant differences were observed in the cellular density between the rotenone treatment group and control group, and even between the 0.2 and 20 µg/mL of rotenone treatment groups (Figure 1). It can be seen that the cells in the control group grew well and consistently, and the cell density kept increasing as time increased (Figure 1A,D,G). After treatment with two concentrations of rotenone, the cell density gradually decreased with time, and this regularity was related to the concentration. Additionally, an interesting phenomenon appeared between the treatment groups. When treated with 0.2 µg/mL of rotenone, the SL-1 cells showed obvious aggregation, and then formed cell clumps (Figure 1B,E,H), whereas under treatment with 20 µg/mL of rotenone, this phenomenon did not appear (Figure 1C,F,I).

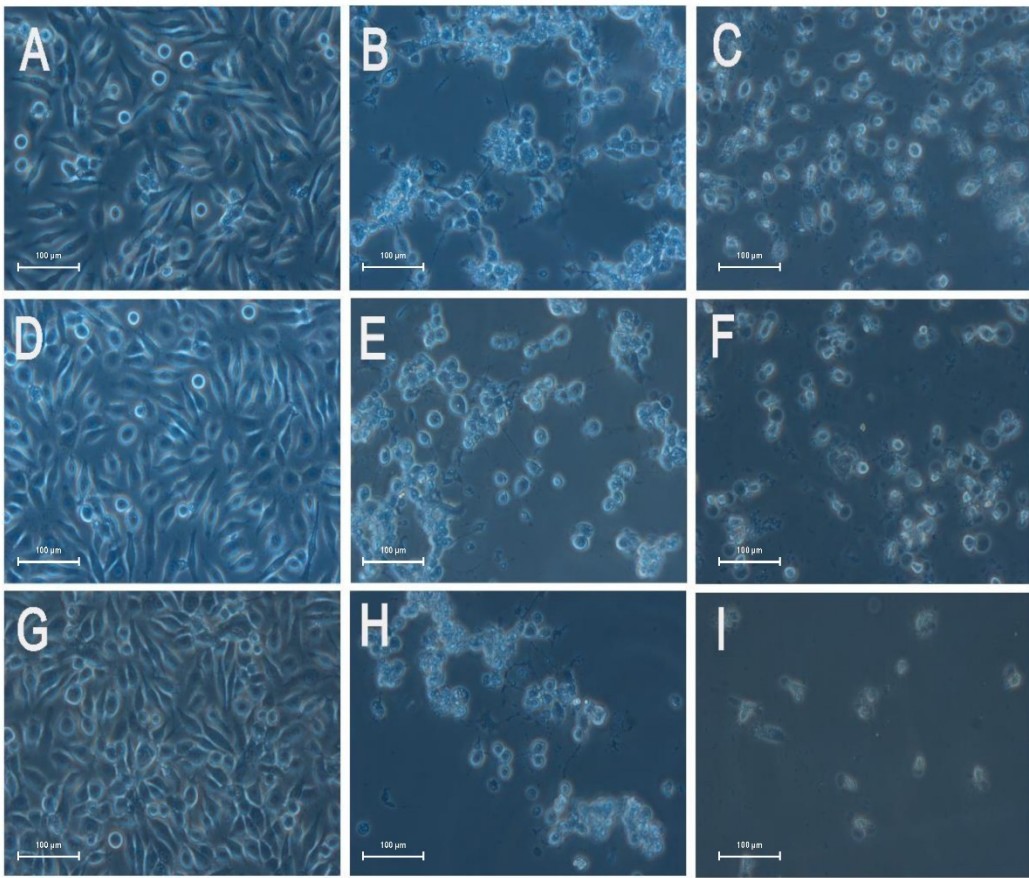

**Figure 1.** Observation of SL-1 cells treated with different concentrations of rotenone at different time points. Control: (**A**) 24 h, (**D**) 48 h, and (**G**) 72 h; 0.2 μg/mL of rotenone: (**B**) 24 h, (**E**) 48 h, and (**H**) 72 h; 20 μg/mL of rotenone: (**C**) 24 h, (**F**) 48 h, and (**I**) 72 h.

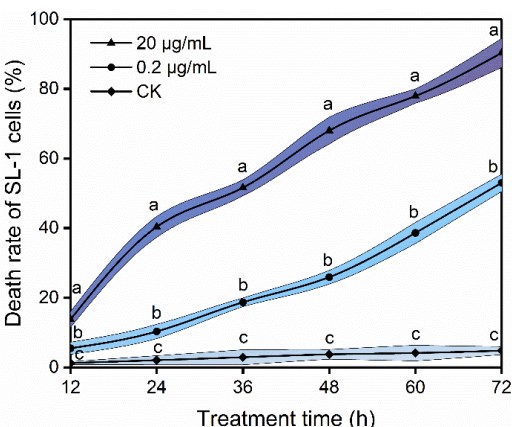

**Figure 2.** Toxicity of rotenone at 20 μg/mL and 0.2 μg/mL to SL-1 cells over 72 h. Data are presented as mean ± S.E (*n* = 3). Different letters at each observation time point indicate significant differences in the death rate of SL-1 cells among treatments at *p* < 0.05 level based on Tukey's HSD test.

To accurately determine the toxicity of rotenone to SL-1 cells at the above two concentrations, the cell death rates within 72 h were measured and shown in Figure 2. A cell death rate of 5.51% was evident when SL-1 cells were treated with 0.2 μg/mL of rotenone for 12 h; the cell death rates gradually increased with time and reached 53.02% at 72 h. When SL-1 cells were treated with 20 μg/mL of rotenone for 12 h, the cell death rate was 13.79%; with the passage of time, the cell death rates continued to increase and reached 90.35% at

72 h. At the same time point, the death rates of SL-1 cells after treatment with 20 μg/mL of rotenone were significantly higher than those of SL-1 cells under 0.2 μg/mL rotenone treatment, and the death rates of SL-1 cells in rotenone treatment groups were significantly higher than those of SL-1 cells in the control group. The results indicated that rotenone showed effective toxicity to SL-1 cells.

### 3.2. Effects of Low Concentration of Rotenone on the Aggregation of SL-1 Cells

In view of the phenomenon of cell aggregation caused by 0.2 μg/mL of rotenone in the above toxicity experiment, the cell aggregation rates and increments in different time periods within 24 h were determined to explore its regularity. As shown in Figure 3A, after 0.2 μg/mL of rotenone treatment, the aggregation rates of SL-1 cells increased with time. The results showed that the aggregation rate of SL-1 cells was 24.71% at 4 h, and increased to 68.14% at 24 h, indicating that treatment time has a significant effect on the aggregation rates of SL-1 cells. The results in Figure 3B showed that the increments of cell aggregation rates were 24.71% and 21.12% at 0–4 h and 4–8 h, respectively, which were significantly higher than the increments of cell aggregation rates in other time periods; these results indicated that SL-1 cells aggregated fastest in the first 8 h after treatment with 0.2 μg/mL of rotenone.

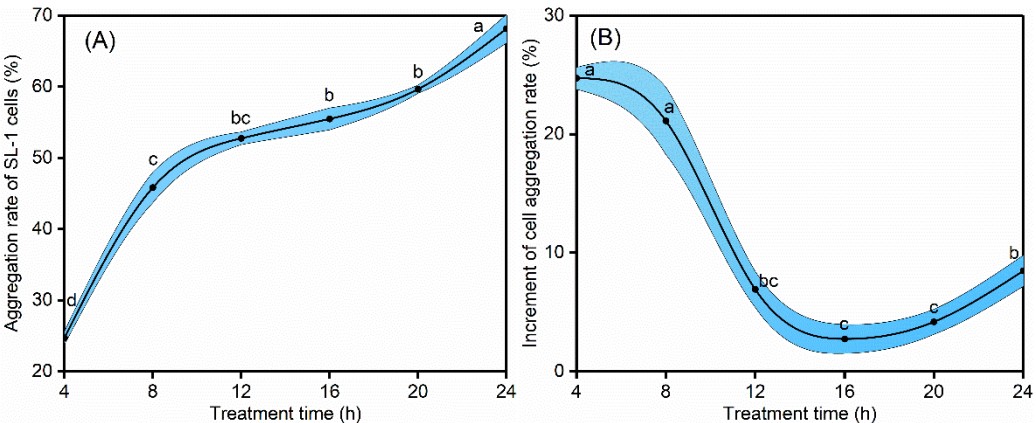

**Figure 3.** Aggregation rate of SL-1 cells (**A**) and increment of cell aggregation rate (**B**) within 24 h after treatment with 0.2 μg/mL of rotenone. Data are presented as mean ± S.E ($n = 3$). Different letters indicate significant differences in the aggregation rate of SL-1 cells or increment of cell aggregation rate in different time periods at $p < 0.05$ level based on Tukey's HSD test.

### 3.3. Effects of Rotenone on the Skeleton of SL-1 Cells

In the toxicity experiment, we observed that SL-1 cells were aggregated under treatment with 0.2 μg/mL of rotenone, while there was no such phenomenon when treated with 20 μg/mL of rotenone. The movement of cells is closely related to the cytoskeleton, thus, the skeletons of SL-1 cells under treatment with the above two concentrations of rotenone for 24 h were further observed. As shown in Figure 4, the cells in the control group had regular morphology with clear skeletons, most of which were spindle-shaped, and the nuclei were intact and clear. After treatment with two concentrations of rotenone, the skeleton of SL-1 cells was broken, and the cells shrank and became irregular. And it can be seen that there was a significant difference between the two rotenone treatment groups. When treated with 0.2 μg/mL of rotenone, an obvious aggregation phenomenon occurred among SL-1 cells; however, treatment with 20 μg/mL of rotenone did not cause cell aggregation. The results further verified that 0.2 μg/mL rotenone treatment caused abnormal movement between SL-1 cells, promoting the aggregation of SL-1 cells, which may lead to a different mode of toxic action.

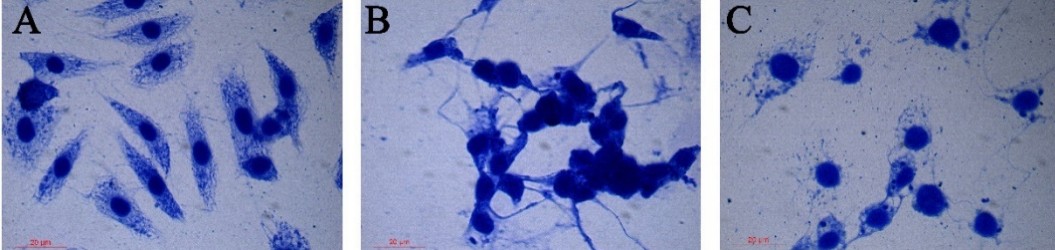

**Figure 4.** Differences in the skeleton of SL-1 cells after 24 h treatment with different concentrations of rotenone. Control: (**A**); 0.2 μg/mL of rotenone: (**B**); 20 μg/mL of rotenone: (**C**).

*3.4. Effects of Rotenone on the Morphology of SL-1 Cell*

To explore whether cell aggregation has an effect on cell morphology, we further observed the cell morphology after rotenone treatment. As shown in Figure 5, the surface cilia of SL-1 cells in the control group were clearly visible, and the cell membrane was normal. After rotenone treatment, the number of surface cilia was significantly reduced, and a significant difference was observed in the cellular membrane after being treated with 0.2 μg/mL of rotenone. It can be seen that after treatment with 0.2 μg/mL of rotenone, the cell membranes were squeezed and pulled due to cell aggregation (Figure 5B), which may affect the function of the cell membrane.

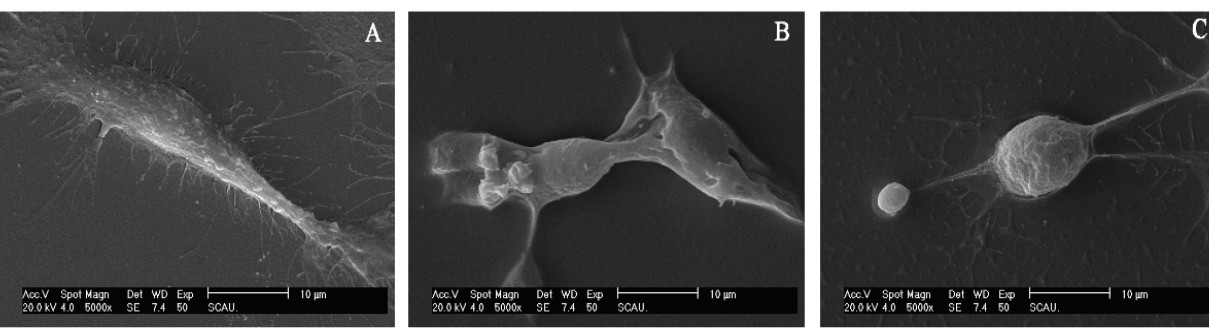

**Figure 5.** Morphological differences in SL-1 cells after 24 h treatment with different concentrations of rotenone. Control: (**A**); 0.2 μg/mL of rotenone: (**B**); 20 μg/mL of rotenone: (**C**).

*3.5. Effects of Rotenone on Membrane Permeability of SL-1 Cells*

In previous experiments, we observed that SL-1 cells aggregated after 0.2 μg/mL of rotenone treatment, resulting in cell membrane squeezing and pulling, possibly affecting the cell membrane function. Thence, we further observed the changes in membrane permeability of SL-1 cells treated with 0.2 and 20 μg/mL of rotenone over 72 h, as shown in Figure 6. It can be seen that only a small number of SL-1 cells in the control group were labeled with PI, indicating that the membrane of SL-1 cells in the control group functioned normally (Figure 1A–C). After treatment with 0.2 μg/mL of rotenone, the number of SL-1 cells labeled with PI was significantly increased, indicating that the permeability of PI to SL-1 cell membrane increased significantly, the function of the cell membrane was affected, and with the increase of treatment time, the damage was greater (Figure 1D–F). However, after treatment with 20 μg/mL of rotenone, the cell membrane function was affected to some extent, but there was no obvious regularity.

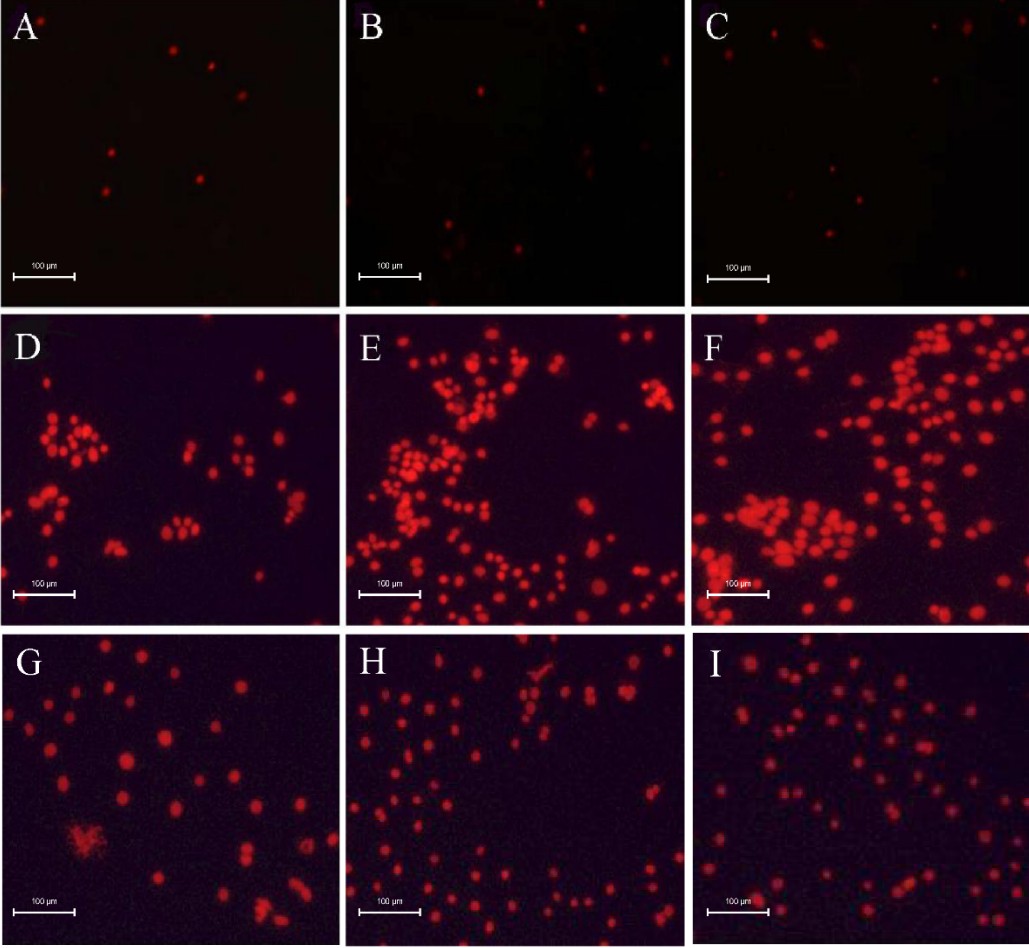

**Figure 6.** Membrane permeability of SL-1 cells over 72 h after treatment with rotenone at 0.2 and 20 μg/mL. Control: (**A**) 24 h, (**B**) 48 h, and (**C**) 72 h; 0.2 μg/mL of rotenone: (**D**) 24 h, (**E**) 48 h, and (**F**) 72 h; 20 μg/mL of rotenone: (**G**) 24 h, (**H**) 48 h, and (**I**) 72 h.

## 4. Discussion

Rotenone has been used for agricultural pest management for decades [26], but most studies on the mechanism of rotenone action have focused on mammalian cells, and relatively few on insect cells. For mammalian cells, previous studies have demonstrated that rotenone inhibits microtubule polymerization and arrests cell cycle progression at mitosis through directly binging to tubulin [27,28], as well as inducing apoptosis in several types of cells by enhancing production of mitochondrial reactive oxygen species [29]. There have been numerous studies reporting cytotoxic activity and mechanism of action of rotenone against several mammalian cells, such as K562, SHSY-5Y, and PC12 [30–32], and the rotenone model of Parkinson's disease [33,34]. As for insect cells, the study by Zhong et al. determined the cytotoxicity of rotenone on the SL-1 and Sf9 cell lines derived from two lepidopteran ovarian cell lines, and the results showed that rotenone could cause the necrosis of SL-1 cells and may be related to mitochondrial depolarization [35], which is consistent with the report that rotenone induces P5 primary neuronal cell necrosis [36].

In this study, we found that rotenone has effective toxicity against *S. litura* larvae and may have some sublethal effects, therefore, the cytotoxic effects of a low concentration (0.2 μg/mL) of rotenone on SL-1 cells was studied using a high concentration (20 μg/mL) of rotenone as a reference point, the results showed that rotenone exhibited an effective toxicity on SL-1 cells (Figure 2), and the toxic effects exhibited by the above two concentrations are different. Interestingly, a phenomenon was initially observed under the inverted microscope. When treated with 0.2 μg/mL of rotenone, SL-1 cells showed obvious

aggregation, whereas this phenomenon did not appear under treatment with 20 µg/mL of rotenone (Figure 1). It can be seen that the aggregation of SL-1 cells was not caused by excessive cell proliferation. Moreover, the aggregation rate of SL-1 cells increased with time, with the fastest aggregation in the first 8 h (Figure 3). Based on the study on the mechanism of toxicity in rotenone models of Parkinson's disease, the results showed that antioxidants could attenuate rotenone-induced cell death; therefore, we speculated that the cell aggregation phenomenon is the result of the toxic effects of reactive oxygen species on cells, because the appearance of carbonylated proteins in the media with cells is the most frequent type of protein modification induced by reactive oxygen species (they are a consequence of mitochondrial rupture). This process is well known and considered irreversible, and means protein degradation, and under the influence of such proteins, aggregation (adhesion) of cells may occur [37]. Moreover, cell aggregation will also bring subsequent chain toxic effects. Subsequently, we further observed changes in the cytoskeleton after rotenone treatment. The results indicated that the cytoskeleton was disrupted after rotenone treatment (Figure 4), which may affect the motility and morphology of SL-1 cells. The cytoskeleton is closely related to cell morphology and movement, and once the cytoskeleton is damaged, the cell cannot continue to maintain its original shape, becomes rounded, and shrinks, and its movement may also be affected [38–40]. The cell aggregation phenomenon caused by 0.2 µg/mL of rotenone may be due to its cytoskeleton being the target site of the pesticide.

As far as it is known, abnormal Leydig cell aggregation in rat fetal testes exposed to di (*n*-butyl) phthalate causes testicular dysplasia [41], and galectin-1 and -3 synthetic inhibitors can selectively regulate homotypic cell aggregation and tumor cell apoptosis, cell aggregation is often considered a pathological phenomenon [42]. Accordingly, we speculated that cell aggregation may lead to different toxic effects. In the present study, the morphology of SL-1 cells treated with 0.2 µg/mL of rotenone was observed by scanning electron microscopy, and it can be seen that the cell membranes were squeezed and pulled due to the cell aggregation, which may affect the function of the cell membrane (Figure 5B). Finally, the membrane function of SL-1 cells after rotenone treatment was measured, and the results showed that the permeability of the cell membrane was increased (Figure 6), indicating that rotenone treatment damaged the SL-1 cell membrane and affected its function. Moreover, rotenone at a concentration of 0.2 µg/mL disrupted SL-1 cell membrane function more strongly compared to rotenone at a concentration of 20 µg/mL. As one of the main defense systems of cells, the cell membrane is a bilayer structure composed of lipids and proteins. Its main function is to maintain normal cell morphology, control the concentration and osmotic pressure of ions inside and outside the membrane, and control the entry and exit of substances inside and outside the membrane [43,44].

## 5. Conclusions

In this study, rotenone exhibited effective toxicity to SL-1 cells, and the effect of the action was different between 0.2 and 20 µg/mL of rotenone treatment. Our results demonstrated that 0.2 µg/mL of rotenone could cause abnormal movement of SL-1 cells leading to cell aggregation, which did not appear under treatment with 20 µg/mL of rotenone, and this phenomenon of cell aggregation would disrupt their membrane morphology and function more strongly compared to the treatment with 20 µg/mL of rotenone. However, the molecular mechanism of 0.2 µg/mL rotenone leading to aggregation of SL-1 cells remains to be further studied. Considering that rotenone has been used to control a variety of pests in agriculture for decades, it is necessary to study its cytotoxic mechanism in pests in more detail for further applications.

**Supplementary Materials:** The following supporting information can be downloaded at: https://www.mdpi.com/article/10.3390/agronomy12112611/s1, Figure S1: Mortality rates of *S. litura* after three days of treating different concentrations of rotenone.

**Author Contributions:** Conceptualization, D.C.; methodology, Q.Z. (Qingpeng Zhang); software, K.X.; validation, Q.Z (Qingpeng Zhang).; formal analysis, S.L.; investigation, S.L.; resources, Z.Z.; data curation, S.L.; Writing—original draft preparation, S.L.; Writing—review and editing, M.M.K.; supervision, D.C.; project administration, Q.Z. (Qiuming Zhu); funding acquisition, Z.Z. All authors have read and agreed to the published version of the manuscript.

**Funding:** This research was funded by the Guangdong Agricultural Science and Technology Innovation Promotion and Agricultural Resources and Ecological Environmental Protection Construction Project, grant number: 2022KJ122, and the Guangzhou Science and Technology Planning Project, grant number: 202102080120.

**Institutional Review Board Statement:** Not applicable.

**Informed Consent Statement:** Not applicable.

**Data Availability Statement:** Not applicable.

**Acknowledgments:** Thanks to Muhammad Zeeshan for his help in revising the manuscript.

**Conflicts of Interest:** The authors declare no conflict of interest.

**Sample Availability:** Samples of the compounds are not available from the authors.

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
