# Peer review of "Low Concentration of Rotenone Impairs Membrane Function of Spodoptera litura Cells by Promoting Their Aggregation"

_agronomy, doi:10.3390/agronomy12112611_

Round 1

Reviewer 1 Report (New Reviewer)

The work by Lin and colleagues shows morphological changes in litura ovarian cells following exposure to sublethal and near lethal concentrations of rotenone, a very well known secondary metabolite which is highly toxic to insects and mammals. The results show clearly detrimental effects, similar to what has been described in other works. I have found some details in the ms that need to be revised, though, which hopefully will improve the quality of the present work.

Title

Please change it to something less descriptive, for example: Effect of rotenone on ovarian s. litura cells is detrimental at even very low concentrations.

Abstract

L14-15 need to be rewritten, they are hard to understand.

Keywords

Please do not include words mentioned in the title of the ms

Introduction

The last paragraph must be removed, since it is part of the results and discussion. Just mention the objectives of your work instead.

Materials and methods

Please rearrange the rotenone and CBB purchases so you don’t repeat that they were acquired from sigma.

L253: How did you measure cell density for your experiments?

L257: Give a brief description of the toxicity tests. Also mention why you chose 20 and 0.2 ug/ml for the following experiments.

L264-265: I dont understand how you exposed the litura cell culture to rotenone, please give more details on this.

L273: this calculation and the following one correspond to percent, not rate.

Author Response

Reviewer 2 Report (New Reviewer)

Add a sentence in the manuscript introduction explaining what SL-1 cells are. Also explain the biological significance of the aggregation of these cells. I believe this will help the reader to understand the results and conclusions presented by the manuscript.

Author Response

Reviewer 3 Report (New Reviewer)

-          Line 242 : (4.1. Chemicals, Cell Line and Culture Conditions) it is necessary to give more details on Spodoptera litura culture conditions (Temperature, humidity, photoperiod…).

-          Line 256: How to justify the choice of the second instar of Spodoptera litura (The toxicity experiment of rotenone against the 2nd instar larvae).

-          Why you have not tested the effect of retonone on pupae and adults?

-          Line 257:Why did you calculate mortality after three days? (the mortality rate after three days of treatment was calculated using the method of Lin et al. [42]).

-          It is preferable to calculate the standard deviations and integrate them for figures 2 and 3.

-          The materials and methods part must be placed before the result part.

Author Response

Reviewer 4 Report (New Reviewer)

I recommend the manuscript to be accepted after revision done. Therefore, I propose:

1) Page 2, lines 53-6: There is no specific action of rotenone, e.g. action on a specific enzyme, ion channel, etc.

2) The presented results and discussions lack specific data on the action of rotenone. The information about the change in cell morphology (line 228) or the effect on the cytoskeleton (line219) is too general.

3) Point 2.5: In figure 6 there is no difference in the cell staining and the description does not explain the obtained results.

4) Points 2.3 i 2.4:  It was not stated what percentage of the cells had changed shape.

5) In the description of the figures, abbreviations should not be used.

6) Figure 2: The markers are poorly visible.

7) Page 8, lines 258: Typo at 2.

8) Sort key-words alphabetically.

Round 2

Reviewer 4 Report (New Reviewer)

I recommend the manuscript titled "Low Concentration of Rotenone Promotes the Aggregation of Spodoptera litura Cell (SL-1) and Impairs Its Membrane Function" to be accepted for publication.

Author Response

Thank you for your affirmation. We have revised the grammar of the manuscript.

This manuscript is a resubmission of an earlier submission. The following is a list of the peer review reports and author responses from that submission.

Round 1

Reviewer 1 Report

The manuscripts titled “Low Concentration of Rotenone Promotes the Aggregation of 2 SL-1 Cell and Impairs Its Membrane Function” present 1 molecule denominated rotenone. The work presents from the cytotoxicity effects of rotenone on insect in two concentrations (low and high). Good manuscript.

Reviewer 2 Report

The study documents some interesting observations on the behavior of SL-1 cells exposed to two different concentrations of rotenone, so in terms of basic cytotoxicity/cell biology, rotenone may be a useful tool.  However, this may have nothing to do with the insecticidal mode-of-action (that which kills insects) of rotenone.

To tie the present data to the insecticidal mode-of-action, the investigators need to compare their in vitro results with SL-1 cells to in vivo toxicity to Spodoptera litura larvae via ingestion.  What is the normal time-to-kill for rotenone in noctuid larvae?  In Figure 2, the LT50 for the high concentration is approximately 36 hr, and about 72 hr for the low concentration.  The authors need to run a bioassay on larvae at doses producing comparable internal concentrations (based on body volume) to see if such doses produce mortality.  Alternatively, there may be published data on rotenone toxicity in lepidopteran larvae that can serve to address this key question.

While studies of cytotoxicity are interesting from a basic viewpoint, they may or may not be relevant to insecticidal mode-of-action.  If not, then the discussion of rotenone as an insecticide should simply be a side note, rather than an important context for the present results.

Reviewer 3 Report

After reading and analyzing the manuscript entitled: Low Concentration of Rotenone Promotions the Aggregation of SL-1 Cell and Impairs It Membrane Function by Sukun Lin, Kaijie Xu, Li Zhang, Muhammad Musa Khan, Yuzhe Chen, Suqing Huang, Dongmei Cheng and Zhixiang Zhang. The manuscript is very clear in terms of objective and focus, with clear figures and scale of figures. I suggest that SL-1 Cell should be replaced by Spodoptera litura Cell (SL-1). Remove Spodoptera litura from keywords.

The manuscript is an evaluation of rotenone as an insecticide, especially using different concentrations, evaluating toxicity to SL-1 cells. I consider the strong point of the manuscript to evaluate the cytotoxicity of rotenone in SL-1 cells with figures that show, in fact, what occurred. Cytotoxicity studies are just one way to show the mode of action of rotenone and were chosen by the authors. It is not the only one, but within the proposal and in view of the objectives listed, applied methodology, and observed results, I consider the conclusions pertinent. Another aspect that leads me to indicate the acceptance of the manuscript is the theoretical basis made, as the discussion around the theme is current, bringing with it scientific solidity to the manuscript and greater interest in the scientific community. As a suggestion it is worth pointing new paths to the authors, without considering this a defect in the manuscript, would be tested for rotenone by ingestion, topical application accompanied by histological analyses, and even evaluation of the biological characteristics of the insect. This can bring the authors real modifications in the use of rotenone on the digestive system, coating, and even nervous.

Reviewer 4 Report

The manuscript is interesting, but I have some doubts. The authors note: "The cell aggregation phenomenon caused by 0.2 μg/mL rotenone may be due to its cytoskeleton being the target site of the pesticide." What are the elements of evidence for affirmation? On the other hand, why does this not occur at a higher dose... The authors should make a proposal or theorize about this. Why didn't they use an intermediate concentration of rotenone?